# LncRNA WDR11-AS1 Promotes Extracellular Matrix Synthesis in Osteoarthritis by Directly Interacting with RNA-Binding Protein PABPC1 to Stabilize SOX9 Expression

**DOI:** 10.3390/ijms24010817

**Published:** 2023-01-03

**Authors:** Huang Huang, Jidong Yan, Xi Lan, Yuanxu Guo, Mengyao Sun, Yitong Zhao, Fujun Zhang, Jian Sun, Shemin Lu

**Affiliations:** 1Department of Biochemistry and Molecular Biology, Institute of Molecular and Translational Medicine, School of Basic Medical Sciences, Xi’an Jiaotong University Health Science Center, Xi’an 710061, China; 2Department of Human Anatomy, Histology and Embryology, School of Basic Medical Sciences, Xi’an Jiaotong University Health Science Center, Xi’an 710061, China; 3Key Laboratory of Trace Elements and Endemic Diseases, National Health Commission, School of Public Health, Xi’an Jiaotong University Health Science Center, Xi’an 710061, China

**Keywords:** osteoarthritis, inflammation, lncRNA WDR11-AS1, PABPC1, SOX9, chondrocyte

## Abstract

Osteoarthritis (OA) is a degenerative disease of articular cartilage that is mainly characterized by chronic and mild inflammation of the joints. Recently, many studies have reported the crucial roles of long noncoding RNAs (lncRNAs) in OA as gene transcriptional regulatory factors, diagnostic biomarkers, or therapeutic targets. However, the exact mechanisms of lncRNAs in the regulation of OA progression remain unclear. In the present study, the lncRNA WDR11 divergent transcript (lncRNA WDR11-AS1) was shown to be downregulated in osteoarthritic cartilage tissues from patients, and to promote extracellular matrix (ECM) synthesis in osteoarthritic chondrocytes with knockdown and overexpression experiments. This function of lncRNA WDR11-AS1 was linked to its ability to interact with the polyadenylate-binding protein cytoplasmic 1 (PABPC1), which was screened by RNA pulldown and mass spectrometry analyses. PABPC1 was discovered to bind ECM-related mRNAs such as *SOX9*, and the inhibition of PABPC1 improved the mRNA stability of *SOX9* to mitigate OA progression. Our results suggest that lncRNA WDR11-AS1 has a promising inhibitory effect on inflammation-induced ECM degradation in OA by directly binding PABPC1, thereby establishing lncRNA WDR11-AS1 and PABPC1 as potential therapeutic targets in the treatment of OA.

## 1. Introduction

Osteoarthritis (OA) is a prevalent whole-joint disorder that is characterized by the progressive destruction of articular cartilage (AC), subchondral bone remodeling, osteophyte formation, and synovial inflammation [1]. Chronic and mild inflammation plays a leading role in tissue degradation and remodeling during the progression of OA [2,3]. Chondrocytes are the only cell type responsible for the synthesis of the extracellular matrix (ECM) in cartilage. Increasing levels of proinflammatory cytokines, including interleukin (IL)-1β, tumor necrosis factor (TNF)-α, and IL-6, accelerate anabolic and catabolic dysregulation in chondrocytes [4]. Since therapies aimed at these cytokines have limited positive effects and harmful side effects [5], we still need to determine the precise mechanisms involved in the pathogenesis of OA. Without changing the DNA sequence, epigenetic mechanisms such as alterations in gene expression may provide molecular targets for revealing OA progression.

Long noncoding RNAs (lncRNAs) are large epigenetic modulators. Through binding microRNAs and proteins, lncRNAs participate in modulating ECM degradation, chondrocyte viability, the immune response, angiogenesis, and lipid metabolism during OA pathogenesis [6]. LncRNA ARFRP1, DANCR, FOXD2-AS1, and HOTAIR regulate inflammatory responses through cytokines, such as IL-1β and TNF-α, in the progression of OA [7]. LncRNA SNHG15 works like a sponge for miR-141-3p and miR-7, and its overexpression alleviates IL-1β-induced ECM imbalance [8,9]. Downregulated linc00623 serves as a competitive endogenous inhibitor for miR-101, and promotes ECM deterioration in chondrocytes exposed to IL-1β [10]. Our previous RNA sequencing data [11] identified a lncRNA, WDR11 divergent transcript (lncRNA WDR11-AS1), that was expressed at low levels in OA patient cartilage compared to normal cartilage tissue. LncRNA WDR11-AS1 is a natural noncoding antisense transcript of WDR11 located at chromosome region 10q26.12. The lncRNA WDR11-AS1 has been identified as a survival-associated lncRNA in glioblastoma [12], and linked to genetic variation in neurodegeneration [13]. However, the biological function and molecular mechanism of lncRNA WDR11-AS1I remain unclear in diseases, including OA.

RNA-binding proteins (RBPs) regulate RNA synthesis, modifications, alternative splicing, subcellular localization, stability, and translation. Many studies have proposed that RBPs have an important role in inflammation and cell metabolism in OA [14]. Polyadenylate-binding protein cytoplasmic 1 (PABPC1) is an abundant conserved eukaryotic RBP that specifically recognizes and binds poly(A)/(A)-rich sequences [15]. PABPC1 mediates the transcript level of immune- and inflammation-related genes by alternative splicing in human gastric adenocarcinoma [16]. The upregulated ADAM15 selectively binds to PABPC1 to trigger chondrocytic cell adhesion [17]. In addition, researchers have noted that PABPC1 contributes to mRNA stability by interacting with long noncoding RNAs (lncRNAs) [18,19]. However, the expression and underlying mechanisms of interaction between PABPC1 and lncRNA WDR11-AS1 remain unknown in OA progression.

In this study, we aimed to determine the function of inflammatory cytokine-induced lncRNA WDR11-AS1 and its binding protein PABPC1 in OA pathogenesis and their underlying mechanisms. The expression levels of lncRNA WDR11-AS1 and PABPC1 in osteoarthritic cartilage were analyzed by bioinformatics and human cartilage tissue analysis. We also investigated the effects of lncRNA WDR11-AS1 and PABPC1 on ECM synthesis in chondrocytes. PABPC1 interacted not only with lncRNA WDR11-AS1, but also with ECM-related genes. We further explored the underlying mechanisms of how lncRNA WDR11-AS1 and PABPC1 affect ECM anabolism in osteoarthritic chondrocytes. All these findings suggest a novel molecular mechanism as a potential target for OA.

## 2. Results

### 2.1. LncRNA WDR11-AS1 Is Decreased in Osteoarthritic Cartilage Tissues from OA Patients and Chondrocytes Stimulated with Proinflammatory Cytokines

To investigate whether lncRNA WDR11-AS1 contributes to the pathogenesis of OA, we analyzed the expression of WDR11-AS1 in osteoarthritic cartilage using the Genevestagaor database, in which data were collected from E-MTAB-4304, E-MTAB-6266, E-MTAB-7313, GSE117999, GSE114007, and GSE110606. Then, we found that lncRNA WDR11-AS1 expression was lower in osteoarthritic cartilage tissues than in nonosteoarthritic cartilage tissues from patients (Figure 1A). To verify this result, we obtained and extracted RNA from cartilage samples of undamaged regions or damaged regions from OA patients. RT-qPCR analysis revealed that lncRNA WDR11-AS1 was downregulated in damaged regions compared with the undamaged regions of the same patients (Figure 1B).

Inflammation, such as proinflammatory factors, plays an important role in the progression of OA. To investigate whether alterations in lncRNA WDR11-AS1 expression are affected by inflammation, we pretreated primary chondrocytes and SW1353 cells with IL-1β and TNF-α at different times and concentrations. We found that the levels of lncRNA WDR11-AS1 decreased in a time- and dose-dependent manner in response to IL-1β and TNF-α stimulation in primary chondrocytes (Figure 1C,D). The same trend was also observed in SW1353 cells (Figure 1E,F). These results indicate that lncRNA WDR11-AS1 expression was reduced in osteoarthritic cartilage, as well as proinflammatory cytokine-treated chondrocytes.

### 2.2. Effects of LncRNA WDR11-AS1 Knockdown and Overexpression on ECM Degradation in OA

ECM degradation is the main cause of cartilage loss in OA. To explore whether the dysregulation of lncRNA WDR11-AS1 participates in ECM degradation, we transfected siRNA and shRNA specifically targeting endogenous WDR11-AS1 into SW1353 cells and primary chondrocytes, respectively. The transfection efficiency of the three siRNAs against lncRNA WDR11-AS1 in SW1353 cells was confirmed using RT-qPCR (Figure 2A). According to the interference ability, siRNA-1 and siRNA-2 were selected to further explore whether downregulated lncRNA WDR11-AS1 could affect the protein levels of ECM in chondrocytes. After the inhibition of lncRNA WDR11-AS1, the protein levels of COLⅡ and ACAN were decreased, and MMP3 was increased (Figure 2B). The protein levels of MMP13, ADAMTS4, and ADAMTS5 revealed no significant change (Figure 2B). To determine whether downregulated lncRNA WDR1-AS1 can be involved in ECM degradation under an inflammatory setting, we transfected cells with the siRNA pool (a combination of siRNA-1 and siRNA-2) or shRNA, and cotreated them with IL-1β (10 ng/mL) or TNF-α (10 ng/mL) for 24 h. In the presence of the siRNA pool and TNF-α treatment, the protein levels of COLⅡ were further decreased, and MMP3 and ADAMTS5 levels were increased (Figure 2C).

The above results were further confirmed in primary chondrocytes. Inhibition of lncRNA WDR11-AS1 alone resulted in the downregulation of COL II and ACAN protein expression, with the same trend as for IL-1β or TNF-α treatment (Figure 2D). ADAMTS5 expression was increased in the single knockdown group as well as in the knockdown plus IL-1β treatment group, but TNF-α treatment decreased the expression after knockdown (Figure 2E). Increased expression of MMP13 was observed only in the presence of cotreatment with knockdown of lncRNA WDR11-AS1 and IL-1β (Figure 2E). Collectively, these results suggest that the knockdown of lncRNA WDR11-AS1 resulted in a reduction in ECM synthesis and an increase in matrix catabolism in chondrocytes, even when responding to inflammatory stimulation.

Considering that the reduction in lncRNA WDR11-AS1 led to chondrocyte ECM degradation, we speculate that its overexpression could have a protective role in OA. SW1353 cells were transfected with the lncRNA WDR11-AS1 overexpression plasmid, and its expression was increased by 708.8-fold compared to that of the scramble control (Figure 3A). In SW1353 cells, overexpression of lncRNA WDR11-AS1 increased COLII levels, even in the presence of TNF-α (10 ng/mL) stimulation (Figure 3B). Levels of ACAN were increased, and MMP13 was decreased only with lncRNA WDR11-AS1 overexpression alone (Figure 3B,C). Levels of MMP3 were decreased only when lncRNA WDR11-AS1 overexpression and TNF-α were both present (Figure 3C).

Furthermore, we transfected primary chondrocytes with the adenovirus overexpressing lncRNA WDR11-AS1, and simultaneously stimulated them with IL-1β (10 ng/mL) or TNF-α (10 ng/mL). Levels of COLII increased only in response to lncRNA WDR11-AS1 overexpression, whereas treatment with IL-1β and TNF-α was ineffective (Figure 3D). Levels of ACAN increased only in the presence of lncRNA WDR11-AS1 overexpression and TNF-α stimulation (Figure 3D). In contrast, the levels of MMP3 decreased in response to lncRNA WDR11-AS1 overexpression with IL-1β or TNF-α stimulation (Figure 3E). MMP13 expression was not affected (Figure 3E). Taken together, the data suggest that lncRNA WDR11-AS1 partially protected ECM degradation in chondrocytes, even under inflammatory stimulation.

### 2.3. LncRNA WDR11-AS1 Interacts with PABPC1 Directly but Has No Effect on PABPC1 Expression

Many studies have reported that lncRNAs can exert their function by interacting with proteins to regulate the expression of target genes [20]. To elucidate the mechanism by which lncRNA WDR11-AS1 regulates ECM degradation, we performed an RNA pulldown assay following mass spectrometry to identify potential proteins associated with WDR11-AS1 in SW1353 cells (Figure 4A and Appendix A). From these proteins, we found that the PABPC1 protein was enriched in biotinylated sense WDR11-AS1 precipitates compared with biotinylated antisense WDR11-AS1 precipitates (Appendix A). Therefore, we assumed that PABPC1 potentially interacted with WDR11-AS1. We then performed an RNA pulldown assay using biotinylated sense WDR11-AS1 and biotinylated antisense WDR11-AS1 to test whether PABPC1 could specifically bind to WDR11-AS1, as revealed by mass spectrometry. As expected, immunoblotting showed that PABPC1 existed in WDR11-AS1 captured precipitates rather than antisense transcripts (Figure 4B). To verify the interaction between WDR11-AS1 and PABPC1, we performed an RIP assay to test whether endogenous WDR11-AS1 could bind PABPC1 in SW1353 cells. The results show that lncRNA WDR11-AS1 was significantly enriched in the precipitate captured by the PABPC1 antibody compared with the IgG control (Figure 4C). These data demonstrate that WDR11-AS1 directly bound to PABPC1 in SW1353 cells.

Studies report that lncRNAs can interact with PABPC1 and regulate PABPC1 protein expression [18,21]. Given the interaction between WDR11-AS1 and PABPC1, we speculated whether WDR11-AS1 regulates PABPC1 protein expression. Then, we predicted the subcellular localization of WDR11-AS1 using the lncATLAS website (http://lncatlas.crg.eu/ (accessed on 24 May 2020)). We found that lncRNA WDR11-AS1 was predominantly localized to the nucleus in H1.hESC and HepG2 cell lines (Figure 4D). This result was further confirmed by RT-qPCR of the nuclear and cytoplasmic RNA fractions from human primary chondrocytes (Figure 4E). The results indicate that lncRNA WDR11-AS1 may have transcriptional regulatory functions. However, PABPC1 protein levels were not altered by knockdown or overexpression of lncRNA WDR11-AS1 in primary chondrocytes (Figure 4F,G). These findings demonstrate that lncRNA WDR11-AS1 localized in the nucleus, bound to PABPC1, and was not involved in the regulation of PABPC1 expression in chondrocytes.

### 2.4. PABPC1 Is Highly Expressed in Cartilage Tissues from OA Patients and Negatively Correlated with SOX9 and COLII

Abnormal changes and functions of PABPC1 have been reported in cancers and the cellular stress response [22,23,24]. To verify whether PABPC1 participates in OA progression, we analyzed its expression in OA cartilage. After analysis of the Genevestagaor database (E-MTAB-4304, E-MTAB-6266, E-MTAB-7313, GSE117999, GSE114007, and GSE110606), we found that PABPC1 was highly expressed in human osteoarthritic cartilage tissues compared with nonosteoarthritic cartilage tissues (Figure 5A). The mRNA expression level of PABPC1 was significantly upregulated in damaged regions compared with undamaged regions in cartilage from OA patients (Figure 5B). We then explored the protein level of PABPC1 in these two regions. Histopathological images showed a discontinuous and microfractured surface, reduced Safranin O staining, and the upregulation of MMP13 in damaged cartilage regions compared with undamaged regions (Figure 5C,D). However, COLII protein expression was not significantly different between the two regions (Figure 5C,D). After ascertaining the damaged cartilage, we found that PABPC1 was significantly higher in the damaged regions than in the undamaged regions (Figure 5C,D). In addition, we analyzed the correlation of *PABPC1* with synthesis-related *COLII* and *SOX9* mRNAs in samples from OA patients. *SOX9* was negatively correlated with *PABPC1* (Figure 5E, *p* < 0.05, *r* = −0.4601), and *COLII* was also negatively correlated with *PABPC1* (Figure 5F, *p* < 0.001, *r* = −0.6224). Thus, osteoarthritic cartilage tissues exhibited higher PABPC1 mRNA and protein expression, which was negatively correlated with SOX9 and COLII mRNAs.

### 2.5. Inhibition of PABPC1 Ameliorates ECM Degradation by Improving the mRNA Stability of SOX9

To investigate whether PABPC1 contributes to ECM degeneration, we transfected three siRNAs targeting PABPC1 into SW1353 cells. Western blot analysis confirmed that the three siRNAs alone or together efficiently reduced PABPC1 protein levels (Figure 6A). The mRNA level of *PABPC1* was significantly reduced by transfecting a siRNA pool containing all three siRNAs (Figure 6B). Furthermore, downregulation of *PABPC1* increased the mRNA levels of *COLII* and *SOX9* and decreased the mRNA levels of *MMP3*, but had no effect on the mRNA levels of *ACAN* and *MMP13* (Figure 6C).

In addition to binding to lncRNAs, PABPC1 also interacts with and regulates the transcription, stability, and translation of mRNAs [19,25,26]. Knockdown of PABPC1 promotes the mRNA expression of *SOX9* (an important transcription factor for cartilage matrix synthesis) and *COLII* (a downstream target gene of SOX9). Therefore, we hypothesize that PABPC1 may interact with ECM-related mRNAs in chondrocytes. Then, we performed a RIP assay on SW1353 cells to demonstrate the binding of PABPC1 to ECM-related mRNAs. Compared with IgG controls, PABPC1 interacted with *SOX9*, but not *COLII*, directly (Figure 6D). Since PABPC1 plays a dual role in the “tail preservation” and “tail removal” of mRNAs, we verified whether PABPC1 regulates the stability of *SOX9* mRNA. We transfected SW1353 cells with the si-PABPC1 pool, and then inhibited RNA transcription with actinomycin D. Inhibition of PABPC1 was determined to enhance the stability of *SOX9* mRNA (Figure 6E). Next, the effect of PABPC1 on SOX9 protein stability was explored in SW1353 cells. As expected, inhibition of PABPC1 enhanced SOX9 protein expression levels and stability (Figure 6F). These results suggest that PABPC1 bound and stabilized *SOX9* mRNA, and that inhibition of PABPC1 attenuated the degradation of ECM.

### 2.6. The LncRNA WDR11-AS1 Regulates SOX9 and COLII Expression through PABPC1

From our previous study’s results, we concluded that WDR11-AS1 affected the catabolism and synthesis of ECM and that PABPC1 affected the degradation of ECM by regulating the expression of SOX9. Consequently, we aimed to verify whether WDR11-AS1 affected ECM synthesis via the PABPC1/SOX9 axis. Therefore, we knocked down or overexpressed the lncRNA WDR11-A1 in primary chondrocytes and cotransfected PABPC1 siRNAs to observe changes in matrix synthesis. Inhibition of WDR11-AS1 reduced SOX9 and COLII expression, and these effects were abolished upon the inhibition of PABPC1 (Figure 7A,C). In addition, overexpression of WDR11-AS1 increased SOX9 and COLII expression, and these effects were further enhanced upon the inhibition of PABPC1 (Figure 7B,D). These findings suggest that lncRNA WDR11-AS1 regulated SOX9 and COLII expression through PABPC1.

## 3. Discussion

In this study, we elucidated the molecular mechanism by which lncRNA WDR11-AS1 promotes ECM synthesis and reduces ECM catabolism. WDR11-AS1, located in the nucleus and downregulated in OA, protected the ECM from degradation even under inflammatory treatment. By conducting RNA pulldown assays following mass spectrometry, we found that WDR11-AS1 interacted with the RNA-binding protein PABPC1, which enhances the expression of SOX9 and COLII (Figure 8). Our findings provide a novel target to ameliorate osteoarthritic cartilage loss.

OA is a common degenerative joint disease that causes pain and disability. As its underlying mechanisms remain unclear, disease-modifying drugs for OA are still limited. Chondrocytes are the only cells that produce ECM components of cartilage to sustain the normal physiological function of joints [27]. During OA progression, a series of changes in the function of chondrocytes leads to a disruption in the balance between the breakdown and synthesis of the matrix [28]. Interleukin-1β (IL-1β) and tumor necrosis factor-α (TNF-α) have been widely reported to be associated with the inflammatory progression of OA [29], and both are frequently used to mimic the environment that drives cartilage degeneration in vivo [30]. Many lncRNAs act as critical modulators in response to cartilage development and OA progression [31,32]. Our results show that the lncRNA WDR11-AS1 was reduced in both damaged cartilage from OA patients and chondrocytes treated with IL-1β and TNF-α. LncRNA WDR11-AS1 has been reported as a risk gene for neurodegeneration and glioblastoma [12,13]. To date, its biofunction remains unknown in diseases, including OA. In our study, overexpressed WDR11-AS1 protected against ECM degradation in chondrocytes by promoting the synthesis of ECM proteins and reducing catabolism, even in the presence of proinflammatory cytokines. Our results indicate the protective role of WDR11-AS1 in OA.

LncRNAs have been reported to regulate a variety of biological processes, including proliferation, metastasis, apoptosis, cell cycle transition, and cell structural integrity, through their subcellular localization and interactions with DNA, proteins, and miRNAs in cis or trans [33]. In this study, we used RNA pulldown following mass spectrometry, and found that WDR11-AS1 interacted with the RNA-binding protein PABPC1. PABPC1 was previously found to bind to the poly(A) tail of mRNA [34], and in recent years, it has been increasingly reported that it can also bind to lncRNAs and miRNAs [18,19,35]. WDR11-AS1 was localized in the nucleus in primary chondrocytes, and did not affect PABPC1 expression. PABPC1 is a nucleocytoplasmic shuttling protein, and its subcellular distribution can be altered dynamically in response to cellular stress or viral infection [36]. Therefore, PABPC1 can bind to WDR11-AS1, which is located in the nucleus.

The RNA-binding protein PABPC1 is considered to be a vital regulator of cytoplasmic mRNA fate, controlling the rate of mRNA transport, translation, and decay [37]. In our study, we found that PABPC1 bound and reduced the stability of SOX9 mRNA. SOX9 belongs to the SRY-related high mobility group (HMG) box (SOX) family of transcription factors, which are indispensable and essential for cell fate determination [38]. During articular cartilage (AC) development, SOX9 efficiently binds to single or double HMG-box site(s) in DNA, thereby transactivating its target genes, including COLII and ACAN [39], which are the most abundant proteins in the ECM. Previous studies have found that SOX9 can help sustain cartilage matrix stiffness and conduct AC preservation responses in chondrocytes during injury [40,41]. SOX9 expression is influenced by multiple factors and signaling pathways [42,43]. An increasing number of studies have focused on the epigenetic molecules regulating SOX9 mRNA stability. METTL3 mediates the methylation and degradation of SOX9 mRNA, thus suppressing COLII expression [44]. FUS stabilizes SOX9 mRNA and promotes chondrocyte-specific protein expression by associating with lncRNA MM2P [45]. Our data reveal that overexpression of WDR11-AS1 enhanced SOX9 expression through its interaction with PABPC1. Considering that PABPC1 could interact with both WDR11-AS1 (nucleus) and SOX9 mRNA (cytoplasm), further studies are needed to verify whether WDR11-AS1 determines the fate of SOX9 by influencing the subcellular distribution of PABPC1.

In conclusion, we demonstrate that lncRNA WDR11-AS1 protects the ECM from degradation to hinder cartilage loss in OA. WDR11-AS1 is associated with inflammation and exhibits matrix-preserving behavior in chondrocytes. WDR11-AS1 interacts with PABPC1, and the inhibition of PABPC1 promotes SOX9 mRNA stability, and thereby enhances ECM synthesis in osteoarthritic chondrocytes. These results suggest that lncRNA WDR11-AS1 and PABPC1 may serve as novel therapeutic factors for improving ECM degradation in OA treatment. Their interplay and intracellular distribution should ignite future research as therapeutic targets for OA.

Our study has some limitations. First, OA is a heterogeneous disease caused by multiple factors. Inflammatory cytokine-induced degradation of the cellular matrix does not fully mimic the progression of OA in patients. Second, we did not demonstrate the protective effect of WDR11-AS1 on cartilage in a mouse model due to the poor cross-species conservation of lncRNA. Finally, as OA is a whole-joint disease, whether WDR11-AS1 plays a role in other joint tissues (synovium, subchondral bone, and meniscus) remains to be further investigated.

## 4. Materials and Methods

### 4.1. Cartilage Sample Collection

The study was approved by the Biomedical Ethics Committee of Xi’an Jiaotong University (approval ID: No. 2018-1805-2, approval date: 18 December 2018) and performed in line with the Declaration of Helsinki. All patients provided written informed consent before the surgery.

OA articular cartilage samples were collected from 39 patients (30 women and 9 men; aged 52–82 years with a mean age of 67.4 ± 5.8 years; Kellgren–Lawrence grade III–IV), who had undergone total knee arthroplasty surgery at Xi’an Honghui Hospital. According to the relative smoothness and severe damage of the cartilage surface, samples were divided into undamaged and damaged groups.

### 4.2. Chondrocytes Culture

The human articular cartilages were cut into pieces and digested in 0.25% trypsin solution (10 mL; SV30031.01, HyClone, Shanghai, China) for 30 min at 37 °C, followed by 0.2% type II collagenase solution (10–15 mL; 17101015, Gibco, LA, CA, USA) for 8–12 h at 37 °C. The cells were precipitated at 1200 r/min for 10 min. The cell pellets were resuspended in Dulbecco’s modified Eagle’s medium (DMEM)/F12 (10 mL; SH30261.01, HyClone) containing 10% fetal bovine serum (FBS; 10100147, Gibco), 1% penicillin/streptomycin (15140122, Thermo Fisher, West Hills, CA, USA). Only the first passage primary chondrocytes at 80% confluence were used for the following experiments.

The SW1353 cells were purchased from American Tissue Culture Collection. Briefly, the cells were cultured in RPMI-1640 medium (SH30809.01, Hyclone) supplemented with 10% FBS and 1% penicillin/ streptomycin. 

### 4.3. Cells Treatment and Transfection

Human primary chondrocytes and SW1353 cells were treated with IL-1β (10139-HNAE, Sino Biological Inc., Beijing, China) and TNF-α (HG10602-M, Sino Biological Inc.) at 10 ng/mL concentrations at different time points (3 h, 6 h, 12 h, 24 h, 48 h) or different concentrations (1, 5, 10, 20, 50 ng/mL) for 24 h, respectively. The primary chondrocytes and SW1353 were seeded at a density of 3 × 10^5^ and 1.2 × 10^5^ per well, respectively. Following treatment, the cells were collected for subsequent experiments.

The chondrocytes were transfected with siRNAs targeting lncRNA WDR11-AS1 (NR_033850.1), PABPC1, and the negative control siRNA (si-NC) (Genepharma, Shanghai, China). In addition, the cells were transfected with plasmids (TranSheepBio, Shanghai, China) including pEGFP-C1-WDR11-AS1 plasmid (oe-AS1, overexpression of lncRNA WDR11-AS1) and pEGFP-C1 plasmid (NC, NC of lncRNA WDR11-AS1 overexpression). The specific siRNA sequences against lncRNA WDR11-AS1 or PABPC1 that were used are depicted in Appendix A. When the cells reached 80% confluency, Lipofectamine 3000 (L3000015, Thermo Fisher) was used for the transfection of siRNAs (50 nM/well) or plasmids (2 μg/well). The culture medium was replaced after 6 h, and subsequent experiments were conducted after 48 h.

The chondrocytes were transfected with adenovirus-vector-based short hairpin RNA HANBIO (Shanghai, China) targeting lncRNA WDR11-AS1 (sh-AS1, 400 MOI/well), and adenovirus-vector-based full-length RNA targeting lncRNA WDR11-AS1 (Ad-AS1, 400 MOI/well). Cells transfected with the empty adenovirus vectors served as negative controls (sh-NC, Ad-NC).

### 4.4. Histological Analysis

Cartilage samples were fixed in 4% paraformaldehyde (BL539A, Biosharp, Hefei, China) for 48 h. The samples were decalcified with EDTA (E1171, Solarbio, Beijing, China) for 1 month. The paraffin-embedded samples were cut into 4 μm thick sections, deparaffinized, and rehydrated for histological study. Safranin O/Fast Green staining was performed by placing each section in Safranin O solution for 15 min. The sections were then counterstained with Fast Green solution for 5 min.

IHC was performed by treatment of the sections with H_2_O_2_ for antigen retrieval. Next, blocking with 5% bovine serum albumin (BSA) for 30 min at room temperature was performed. The sections were then incubated with primary antibodies at 4 °C overnight. After incubation with biotinylated secondary antibody, protein staining was performed via the ready-to-use SABC-AP (rabbit IgG) Kit (SA1052, BOSTER, Los Angeles, CA, USA). The images were captured using an Olympus BX51 microscope (Olympus Corporation, Tokyo, Japan). Further analysis was performed using Image-Pro^®^ Plus software (version 6.0.0.260, Media Cybernetics Corporation, Baltimore, MD, USA).

The primary antibodies used include anti-COL2A1 (1:50, A00517-1, BOSTER, Wuhan, China), anti-MMP13 (1:100, 18165-1-AP, Proteintech, Wuhan, China), and anti-PABPC1 (1:200, 10970-1-AP, Proteintech). The secondary antibody was goat anti-rabbit IgG (1:500, #72040-63-2, BOSTER).

### 4.5. RNA Extraction and RT-qPCR

For RNA extraction of cartilage tissues and cultured cells, the tissues and cells were homogenized in Trizol reagent (15596-026; Thermo Fisher Scientific) by moderate vortexing. After homogenization, total RNA was extracted following the protocol of Trizol reagent. The complementary DNA (cDNA) from pure RNA (2 μg) was reverse-transcribed using the cDNA Synthesize Kit (K1622; Thermo Fisher Scientific). Fluorescence-based qPCR assay was performed with the SYBR Green system (04913850001, Roche, Shanghai, China) using the qPCR instrument (Agilent Mx3005P, Agilent Tech, Los Angeles, CA USA). The relative gene expression was calculated, normalized to β-ACTIN, and analyzed using the 2^−ΔΔCT^ method. The primers were synthesized by Sangon Biotech (Appendix A).

### 4.6. Western Blot

Whole-cell proteins were extracted from cells by radioimmunoprecipitation assay buffer (P0013C, Beyotime, Beijing, China) with a protease inhibitor cocktail (P1006, Beyotime). The concentration of protein was quantified by the BCA Protein Assay Kit (PA115, TIANGEN, Beijing, China). Equal amounts of protein were separated by 10% sodium dodecyl sulfate–polyacrylamide (SDS-PAGE) gels, and subsequently transferred onto polyvinylidene difluoride (PVDF) membranes (PVH00010, Millipore, Darmstadt, Germany). Next, the membranes were blocked in 5% non-fat milk at room temperature for 2 h. After incubation with primary antibodies at 4 °C overnight, the membranes were washed three times with Tris-buffered saline Tween-20 (TBST). Subsequently, the membranes were incubated for 2 h at room temperature with horseradish-peroxidase-labeled secondary antibodies. The target proteins were detected by ECL Chemiluminescence Kit (P2200, NCM Biotech, Suzhou, China) and scanned using the GeneGnome XRQ System (Syngene, MD, USA). The bands were analyzed using Image J analysis software.

The respective primary antibodies used include anti-COL2A1 (1:400, A00517-1, BOSTER), anti-ACAN (1:500, 13880-1-Ig, Proteintech), anti-MMP3 (1:500, 14351S, CST), anti-MMP13 (1:500, 18165-1-AP, Proteintech), anti-ADAMTS4 (1:500, A2525, ABclonal, Wuhan, China), anti-ADAMTS5 (1:500, A2836, ABclonal), anti-SOX9 (1:1000, ab185966, Abcam, Cambridge, UK), anti-PABPC1 (1:1000, 10970-1-AP, Proteintech), anti-GAPDH (1:5000, 60004-1-Ig, Proteintech), and anti-β Actin (1:1000, 66009-1-Ig, Proteintech). In addition, the secondary antibodies used include donkey anti-mouse IgG (1:10,000, A21202, Thermo Fisher) and goat anti-rabbit IgG (1:10,000, A11036, Thermo Fisher).

### 4.7. Nucleus and Cytoplasm RNA Fractionation

The cytoplasmic and nucleic RNAs were extracted from freshly cultured primary chondrocytes using the PARIS™ Kit (AM1921, Thermo Fisher) according to the manufacturer’s instructions. Following washing with pre-cooled PBS, the cells were fractionated and centrifuged to obtain the supernatant (cytoplasm) and the pellet (nucleus). RNA was extracted from each fraction.

### 4.8. RNA Pulldown Assay

LncRNA WDR11-AS1 and its antisense RNA were synthesized using the HiScribe™ T7 Quick High Yield RNA Synthesis Kit (E2050S, NEB, Los Angeles, CA, USA). According to the manufacturer’s instructions, the RNA pulldown assay was performed by Pierce™ RNA-Protein PullDown Kit (20614, Thermo Fisher). The 3′ end biotinylated RNAs were bound to the beads and incubated with protein lysates at 4 °C for 2 h. After washing and eluting, the pulldown products were subjected to 10% SDS–PAGE gel and silver staining was performed using the Fast Silver Stain Kit (P0017S, Beyotime). Individual gel lanes were separated for mass spectrometry analysis (Bioprofile, Shanghai, China).

### 4.9. RNA Immunoprecipitation (RIP) Assay

An RIP assay was performed using the Magna RIP Kit (#17-700, Millipore) according to the manufacturer’s instructions. Freshly cultured SW1353 cells were harvested and resuspended in RIP lysis buffer. The cell lysate was incubated with magnetic beads conjugated with anti-PABPC1 antibody or rabbit IgG control overnight at 4 °C. The unbound materials were washed away on the following day, and the magnetic beads were incubated with proteinase K. TRIzol reagent was used to isolate RNAs from the extracts. Finally, the relative enrichments of lncRNA WDR11-AS1 and COLII, ACAN, COL10, MMP3, MMP13, SOX6, and SOX9 were determined by RT-qPCR analysis.

### 4.10. RNA Decay Assay

SW1353 cells were seeded in 12-well plates and infected with si-PABPC1 or si-NC for 24 h. Cells were treated with actinomycin D (ActD; A9415, Sigma, Darmstadt, Germany) at a concentration of 5 μg/mL. The cells were collected at 0, 1, 2, 4, 6, and 8 h post treatment. Finally, the TRIzol reagent was used to extract RNA. Further analysis was performed using RT-qPCR.

### 4.11. Protein Stability Assay

SW1353 cells were seeded in six-well plates and infected with si-PABPC1 or si-NC for 48 h. The cells were treated with cycloheximide (CHX; 66-81-9, Sigma) at a concentration of 50 mg/mL. Finally, the cells were harvested at 0, 1, 2, 3, 4, and 5 h post treatment. Further analysis was performed using Western blot analysis. 

### 4.12. Statistical Analysis

Data were analyzed using SPSS 18.0 statistical software (IBM Corp., Armonk, NY, USA), and the data are expressed as the mean ± standard deviation (SD). Data between OA undamaged cartilage and OA damaged cartilage from the same patients were compared using paired *t*-tests. Comparisons between the two groups were performed by Mann–Whitney U test. Correlation analysis was performed by Spearman correlation. A value of *p* < 0.05 was considered to be statistically significant.

## Figures and Tables

**Figure 1 ijms-24-00817-f001:**
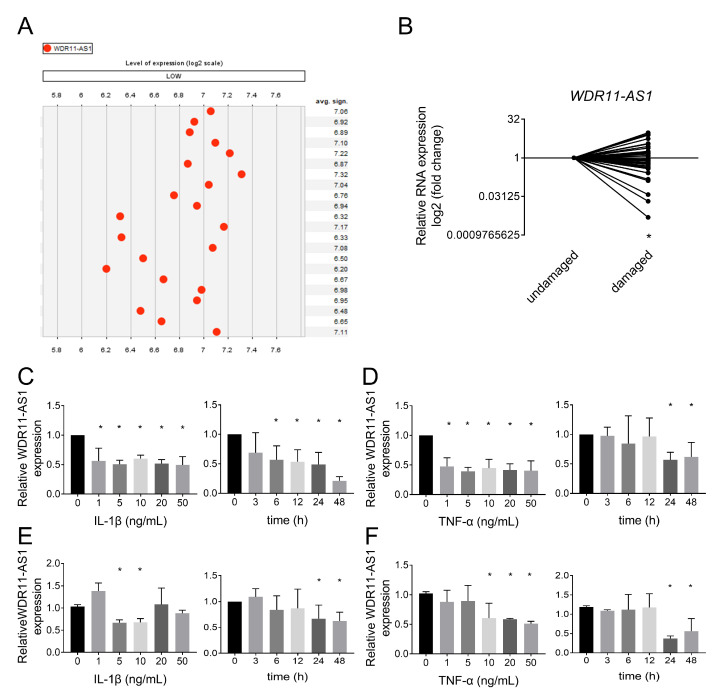
Poor expression of lncRNA WDR11-AS1 in OA cartilage tissues and proinflammatory cytokines stimulated chondrocytes. (**A**) LncRNA WDR11-AS1 levels were extracted from human osteoarthritic cartilage compared with nonosteoarthritic cartilage by the Genevestigator database (E-MTAB-4304, E-MTAB-6266, E-MTAB-7313, GSE117999, GSE114007, GSE110606). (**B**) LncRNA WDR11-AS1 levels were determined by RT-qPCR in cartilage tissues of undamaged and damaged regions in OA patients; *n* = 39. (**C**,**D**) Determination of lncRNA WDR11-AS1 levels in primary chondrocytes stimulated with 10 ng/mL IL-1β (**C**, **right**) or TNF-α (**D**, **right**) for 0, 3, 6, 12, 24, or 48 h and 0, 1, 5, 10, 20, or 50 ng/mL IL-1β (**C**, **left**) or TNF-α (**D**, **left**) for 24 h by RT-qPCR assay. (**E**,**F**) Determination of lncRNA WDR11-AS1 levels in SW1353 cells stimulated with 10 ng/mL IL-1β (**C**, **right**) or TNF-α (**D**, **right**) for 0, 3, 6, 12, 24, or 48 h and 0, 1, 5, 10, 20, or 50 ng/mL IL-1β (**C**, **left**) or TNF-α (**D**, **left**) for 24 h by RT-qPCR assay. * *p* < 0.05, damaged vs. undamaged (**B**), the stimulated group vs. the untreated group (0 ng/mL or 0 h) (**C**–**F**). For (**B**), the lncRNA WDR11-AS1 levels in cartilage tissues were compared using a paired *t*-test. For (**C**–**F**), the experiments were repeated three times. The measurement data are expressed as the mean ± standard deviation, and were analyzed by Mann–Whitney U test.

**Figure 2 ijms-24-00817-f002:**
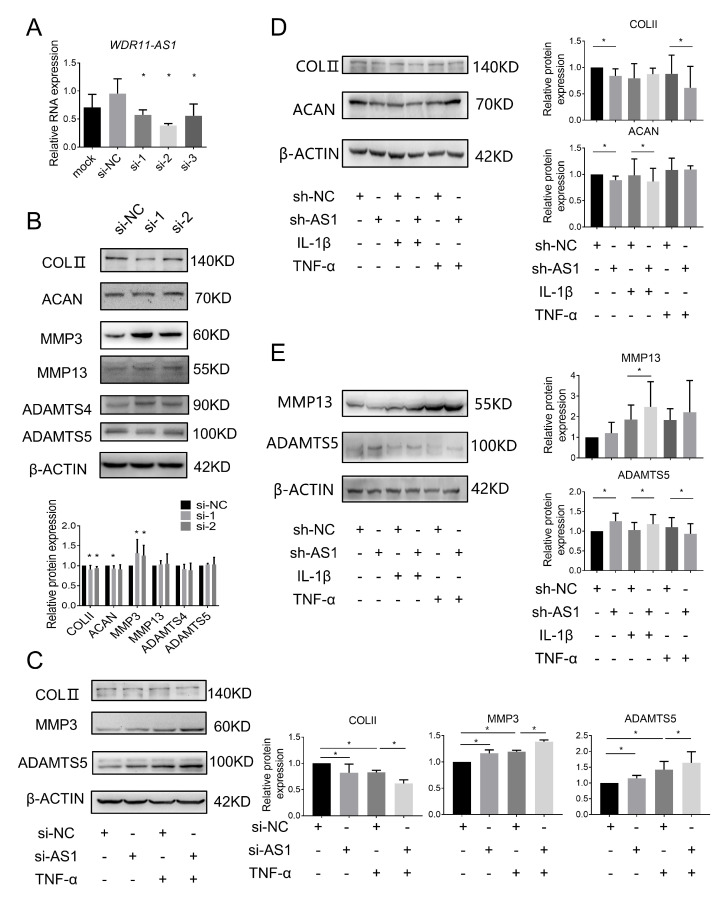
Inhibition of lncRNA WDR11-AS1 promotes ECM degradation in chondrocytes. (**A**) Expression of lncRNA WDR11-AS1 was determined by RT-qPCR in SW1353 cells transfected with three siRNAs (50 nM) targeting endogenous WDR11-AS1. (**B**) Western blot analysis showed COLII, ACAN, MMP3, MMP13, ADAMTS4, and ADAMTS5 protein levels in SW1353 cells after WDR11-AS1 knockdown. (**C**) Western blot analysis showed COLII, MMP3, and ADAMTS5 protein levels in TNF-α (10 ng/mL)-stimulated SW1353 cells after WDR11-AS1 knockdown. (**D**,**E**) Western blot analysis showed COLII, ACAN (**D**), MMP13, and ADAMTS5 (**E**) protein levels in IL-1β or TNF-α (10 ng/mL)-stimulated primary chondrocytes after WDR11-AS1 knockdown. * *p* < 0.05, si-RNA vs. si-NC (**A**,**B**); the measurement data are expressed as the mean ± standard deviation, and were analyzed by Mann-Whitney U test. The experiments were repeated three times.

**Figure 3 ijms-24-00817-f003:**
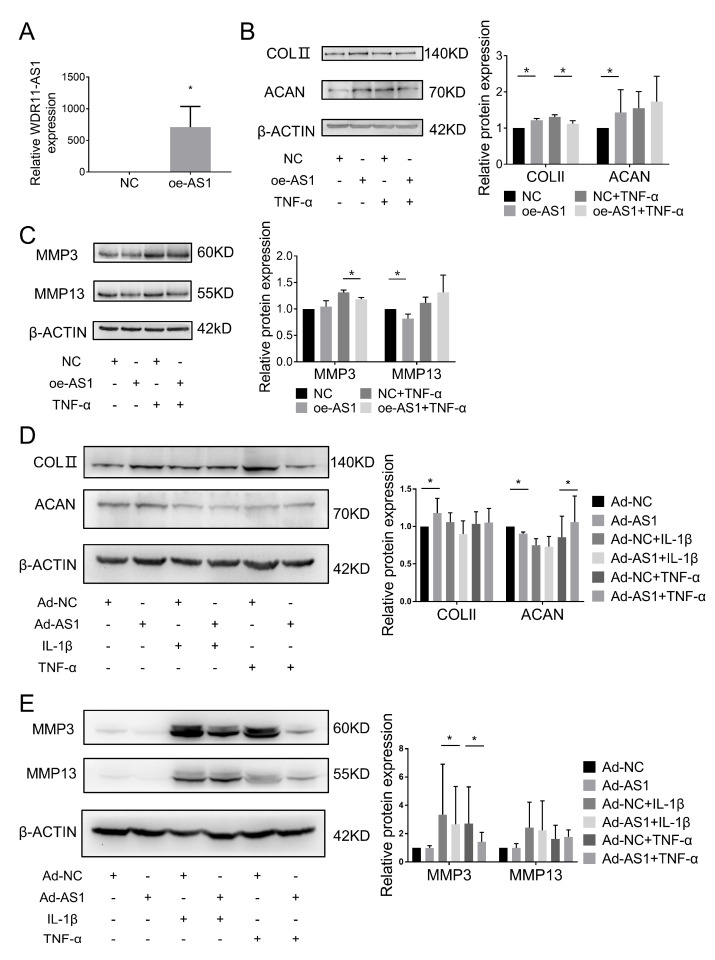
Overexpression of lncRNA WDR11-AS1 inhibits ECM degradation in chondrocytes. (**A**) Expression of lncRNA WDR11-AS1 was determined by RT-qPCR in SW1353 cells transfected with pEGFP-C1-WDR11-AS1 plasmid (oe-AS1, overexpression of lncRNA WDR11-AS1). (**B**,**C**) Western blot analysis showed COLII, ACAN (**B**), MMP3 and MMP13 (**C**) protein in TNF-α (10 ng/mL)-stimulated SW1353 cells after WDR11-AS1 overexpression. (**D**,**E**) Western blot analysis showed COLII ACAN (**D**), MMP3, and MMP13 (**E**) protein levels in IL-1β or TNF-α (10 ng/mL)-stimulated primary chondrocytes after WDR11-AS1 overexpression. * *p* < 0.05, oe -AS1 vs. NC (**A**); the measurement data are expressed as the mean ± standard deviation, and were analyzed by Mann–Whitney U test. The experiments were repeated three times.

**Figure 4 ijms-24-00817-f004:**
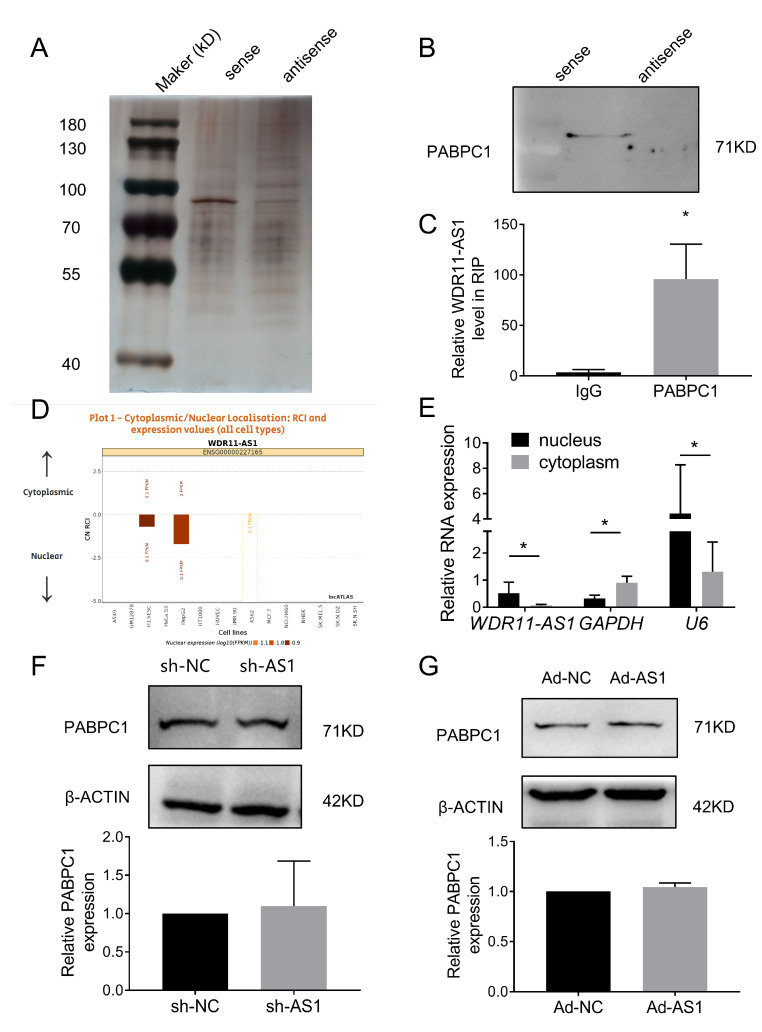
LncRNA WDR11-AS1 binds to PABPC1, but does not alter PABPC1 expression, in chondrocytes. (**A**) Silver-stained gel of WDR11-AS1 interacting proteins isolated from SW1353 cells subjected to RNA pulldown assay using sense or antisense of WDR11-AS1. (**B**) Western blot analysis showed the binding of WDR11-AS1 to PABPC1 from RNA pulldown assay. (**C**) RT-qPCR from RIP analysis showed WDR11-AS1 in PABPC1 immunoprecipitates and IgG (control) from SW1353 lysates. (**D**) Prediction of subcellular localization of lncRNA WDR11-AS1 on the lncATLAS website. (**E**) Detection of subcellular localization of lncRNA WDR11-AS1 in primary chondrocytes by fractionation of nuclear and cytoplasmic RNA analysis. (**F**,**G**) Western blot analysis showed PABPC1 protein levels after WDR11-AS1 knockdown (**F**) or overexpression (**G**) in primary chondrocytes. * *p* < 0.05, PABPC1 vs. IgG (**C**); for (**C**,**E**–**G**), the measurement data are expressed as the mean ± standard deviation, and were analyzed by Mann–Whitney U test. The experiments were repeated three times.

**Figure 5 ijms-24-00817-f005:**
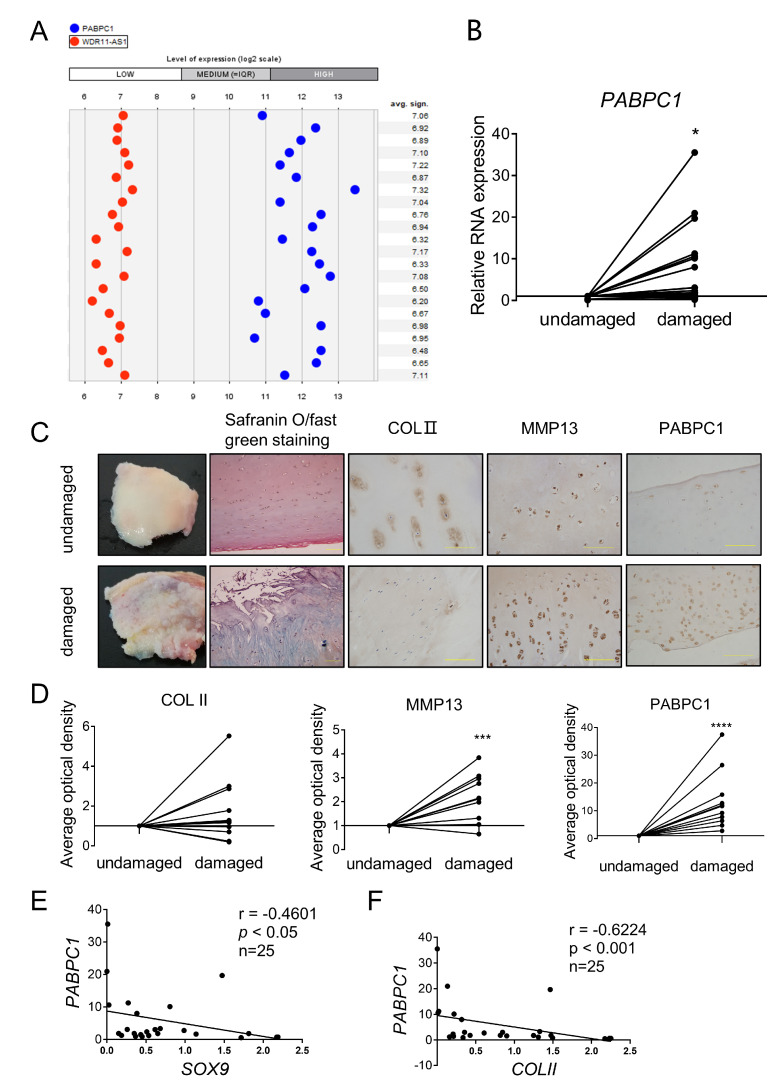
High expression of PABPC1 in OA cartilage tissues. (**A**) Expression levels of *PABPC1* and *WDR11-AS1* in human osteoarthritic cartilage compared with nonosteoarthritic cartilage data from the Genevestigator database (E-MTAB-4304, E-MTAB-6266, E-MTAB-7313, GSE117999, GSE114007, GSE110606). (**B**) PABPC1 mRNA levels were determined by RT-qPCR in cartilage tissues of undamaged and damaged regions in OA patients; *n* = 25. (**C**) Representative photographs of Safranin O/Fast Green staining and immunohistochemistry for COLII, MMP13, PABPC1; 200× magnification; scale bars, 100 μm. (**D**) Quantification of the immunohistochemistry for COLII (*n* = 11), MMP13 (*n* = 10), and PABPC1 (*n* = 11). (**E**) The correlation between SOX9 mRNA and PABPC1 mRNA in OA patients was determined by Spearman correlation analysis. (**F**) The correlation between COLII mRNA and PABPC1 mRNA in OA patients was determined by Spearman correlation analysis. * *p* < 0.05, *** *p* < 0.001, **** *p* < 0.0001, damaged vs. undamaged (**B**,**D**). Data were compared using paired *t*-tests.

**Figure 6 ijms-24-00817-f006:**
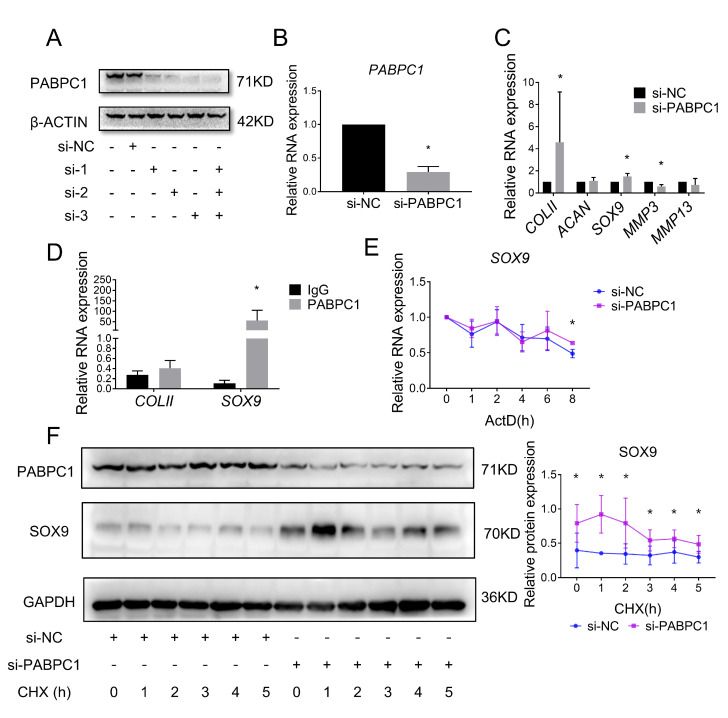
Silencing PABPC1 suppresses cartilage ECM degradation. (**A**) Western blot analysis showed PABPC1 protein levels after transfection of SW1353 cells with three siRNAs (50 nM) targeting endogenous PABPC1 alone or in combination. (**B**) Expression of PABPC1 mRNA levels were determined by RT-qPCR in SW1353 cells after PABPC1 knockdown using siRNA pool (si-PABPC1, a combination of three siRNAs (50 nM)). (**C**) Expression of COLII, ACAN, SOX9, MMP3, and MMP13 mRNA levels were determined by RT-qPCR in SW1353 cells after PABPC1 knockdown. (**D**) RT-qPCR from RIP analysis showed COLII and SOX9 mRNA in PABPC1 immunoprecipitates and IgG (control) from SW1353 lysates. (**E**) RT-qPCR showed that the inhibition of PABPC1 increased the mRNA stability of SOX9. SW1353 cells were treated with actinomycin D (ActD, 5 μg/mL) at different time points 48 h after PABPC1 knockdown. (**F**) Western blot analysis showed that inhibition of PABPC1 increased the protein stability of SOX9. SW1353 cells were treated with cycloheximide (CHX, 50 mg/mL) at different time points 48 h after PABPC1 knockdown. * *p* < 0.05, si-PABPC1 vs. si-NC (**B**,**C**), PABPC1 vs. IgG (**D**), si-PABPC1 vs. si-NC (**E**,**F**) at the same time point are compared. The measurement data are expressed as the mean ± standard deviation, and were analyzed by Mann–Whitney U test. The experiments were repeated three times.

**Figure 7 ijms-24-00817-f007:**
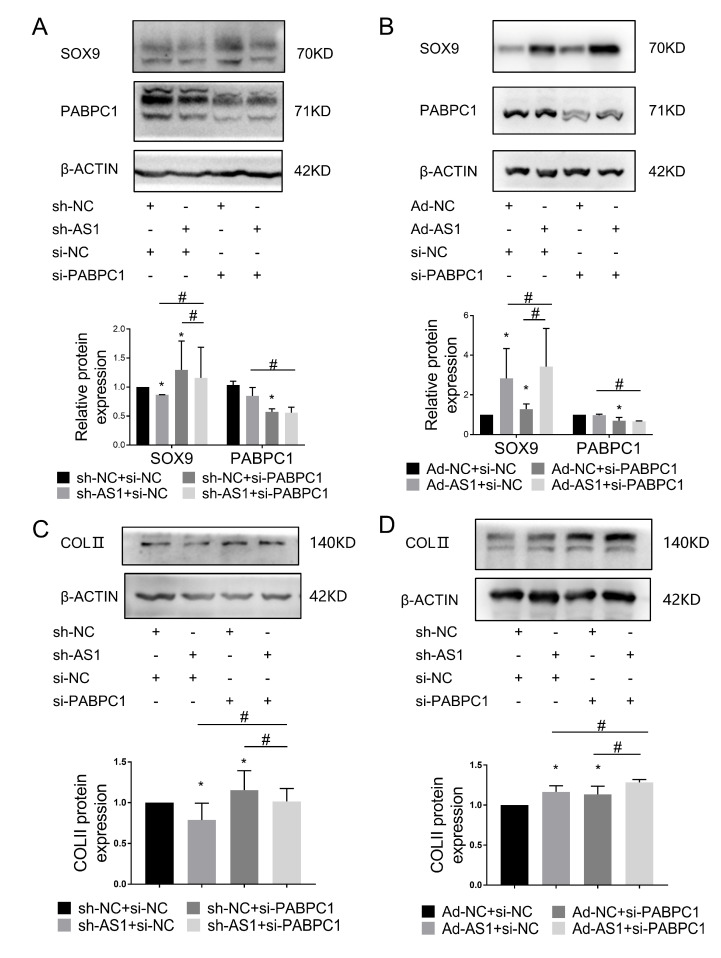
The binding of lncRNA WDR11-AS1 to PABPC1 stabilizes SOX9 expression and induces ECM synthesis in chondrocytes. (**A**) Western blot analysis showed SOX9 and PABPC1 levels in primary chondrocytes reducing WDR11-AS1 with siRNA-mediated PABPC1 knockdown. (**B**) Western blot analysis showed SOX9 and PABPC1 levels in primary chondrocytes overexpressing WDR11-AS1 with siRNA-mediated PABPC1 knockdown. (**C**) Western blot analysis showed COLII levels in primary chondrocytes reducing WDR11-AS1 with siRNA-mediated PABPC1 knockdown. (**D**) Western blot analysis showed COLII levels in primary chondrocytes overexpressing WDR11-AS1 with siRNA-mediated PABPC1 knockdown. * *p* < 0.05, the group vs. the sh-NC+si-NC group; # *p* < 0.05, the two groups under the horizontal line are compared. The measurement data are expressed as the mean ± standard deviation, and were analyzed by Mann-Whitney U test. The experiments were repeated three times.

**Figure 8 ijms-24-00817-f008:**
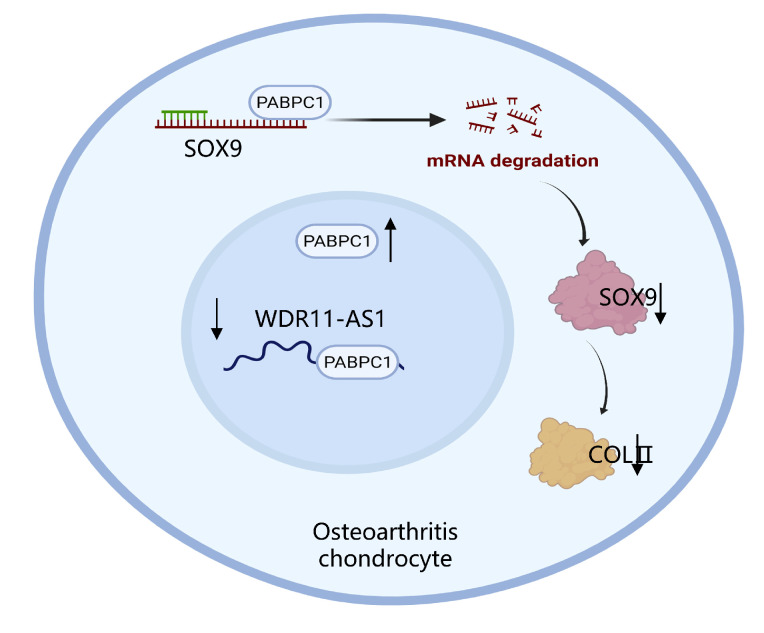
Schematic diagram representing the lncRNA WDR11-AS1 and PABPC1 interaction controlling ECM synthesis in OA chondrocytes. Upregulation of PABPC1 interferes with lncRNA WDR11-AS1 regulation and enhances SOX9 mRNA decay, thereby reducing ECM synthesis.

## Data Availability

The datasets analyzed in the current study are available at https://www.ncbi.nlm.nih.gov/ (accessed on 1 December 2019).

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
