# Peer review of "LncRNA WDR11-AS1 Promotes Extracellular Matrix Synthesis in Osteoarthritis by Directly Interacting with RNA-Binding Protein PABPC1 to Stabilize SOX9 Expression"

_ijms, 2023, doi:10.3390/ijms24010817_

Round 1

Reviewer 1 Report

The article Manuscript (ID ijms-2078600) entitled "LncRNA WDR11-AS1 promotes extracellular matrix synthesis in osteoarthritis by directly interacting with RNA-binding protein PABPC1" is interesting however, some flaws needs to be overcome before its conideration for publication

Introduction:

The part should be improved and associated more updated references. In particular, the section explaining lncRNA is poor in details.

The aim of the study needs to be enriched with more details to better understand authors purposes.

Methods:

Why the authors treated the cultures with two different negative stimuli? Please explain.

The paragraph of real time pcr analysis should be improved.

Results:

In figure 1 the graphs should be modified with the only bars...the authors could delete the spots relatives to the repeated data because they are confounding ( the same for the other figures).

What means “relative expression” if it was expressed in ng/ml? Please explain

Discussion:

In this section, the authors should better stress what all of the findings taken together mean and what is the relevance of the obtained results from the clinical point of view.

Please add the limitations of the study. 

Author Response

Please see the attachment and the revised manuscript.

Reviewer 2 Report

Authors report that lncRNA WDR11-AS1, which is an lncRNA WDR11 divergent transcript, was downregulated in osteoarthritic cartilage tissues. Inhibition of lncRNA WDR11-AS1 promoted ECM degradation in chondrocytes, whereas its overexpression inhibited ECM degradation in the same cells. LncRNA WDR11-AS1 interacted directly with an RNA-binding protein, PABPC1, with little effect on PABPC1 expression in chondrocytes. PABPC1, was highly expressed in OA cartilage tissues and negatively correlated with SOX9 and COLâ…¡. Inhibition of PABPC1 ameliorated ECM degradation by improving the mRNA stability of SOX9. The binding of lncRNA WDR11-AS1 to PABPC1 stabilized SOX9 expression and induced ECM synthesis in chondrocytes, suggesting that  the lncRNA WDR11-AS1 regulates SOX9 and COLâ…¡ expression through PABPC1. The presented data support authors argument and will ignite further research on the detailed molecular mechanisms of OA, including the interplay between WDR11-AS1 and PABPC1, their trafficking or intracellular distribution, and their usefulness as therapeutic targets for OA.

The title can be improved to encompass all aspects of the data and argument.

 typos and errors throughout the manuscript must be addressed before publication. Some of them are listed below.

lines 245-246: PABPC1 mRNA levels were extracted from human osteoarthritic cartilage compared with nonosteoarthritic cartilage -> This sentence can be improved.

lines 281-292 overlap with lines 308-319.

line 375: highcmobility group (HMG) -> high mobility group (HMG)

Author Response

(The authors gave the same response as above.)

Round 2

Reviewer 1 Report

The Authors have modified the manuscript according to reviewer's requests. The manuscript has been improvent and now can be accettable for publication.